# An integrative systematic review on interventions to improve layperson's ability to identify trustworthy digital health information

**Hind Mohamed**[1]*, **Esme Kittle**[1], **Nehal Nour**[1], **Ruba Hamed**[1], **Kaylem Feeney**[1],
**Jon Salsberg**[1], **Dervla Kelly**[1,2]

**1** School of Medicine, University of Limerick, Co. Limerick, Ireland, **2** Health Research Institute, University of limerick, Co. Limerick, Ireland

* hind-awdw@hotmail.com, 20138148@studentmail.ul.ie

**Data Availability Statement:** All relevant data are within the manuscript and its Supporting

## Abstract

Health information on the Internet has a ubiquitous influence on health consumers' behaviour. Searching and evaluating online health information poses a real challenge for many health consumers. To our knowledge, our systematic review paper is the first to explore the interventions targeting lay people to improve their e-health literacy skills. Our paper aims to explore interventions to improve laypeople ability to identify trustworthy online health information. The search was conducted on Ovid Medline, Embase, Cochrane database, Academic Search Complete, and APA psych info. Publications were selected by screening title, abstract, and full text, then manual review of reference lists of selected publications. Data was extracted from eligible studies on an excel sheet about the types of interventions, the outcomes of the interventions and whether they are effective, and the barriers and facilitators for using the interventions by consumers. A mixed-methods appraisal tool was used to appraise evidence from quantitative, qualitative, and mixed-methods studies. Whittemore and Knafl's integrative review approach was used as a guidance for narrative synthesis. The total number of included studies is twelve. Media literacy interventions are the most common type of interventions. Few studies measured the effect of the interventions on patient health outcomes. All the procedural and navigation/ evaluation skills-building interventions are significantly effective. Computer/internet illiteracy and the absence of guidance/facilitators are significant barriers to web-based intervention use. Few interventions are distinguished by its implementation in a context tailored to consumers, using a human-centred design approach, and delivery through multiple health stakeholders' partnership. There is potential for further research to understand how to improve consumers health information use focusing on collaborative learning, using human-centred approaches, and addressing the social determinants of health.

Information files. The search strategy is detailed in the methods and appendices to allow the search to be replicated. The data extraction sheet is also available in the appendix to allow for repeated exploration of the data.

**Funding:** The author(s) received no specific funding for this work.

**Competing interests:** The authors have declared that no competing interests exist.

## Author summary

Access to health information allows people to participate actively in their health care. The internet is an increasingly popular way for people to obtain health information. However, several barriers prevent people from using such information, such as a lack of knowledge and skills to find, evaluate, and use online health information (online health literacy). Our review explored the approaches to improve laypeople's online health literacy. Only twelve studies met the inclusion criteria for this review. We found approaches focusing on teaching online health literacy skills are the most common. In addition, learning computer/internet skills alongside with media literacy is an effective technique, commonly used amongst older people. Lack of computer/internet experience is a common barrier identified in our paper to participating in training. We also observed the importance of providing a convenient training location and involving laypeople in the training design to facilitate its use. Program facilitators could play a critical role in guiding laypeople how to search online health information. Our study provides novel insight into how to improve consumers' health information use and how to provide equity based educational techniques tailored to public needs.

## Introduction

The active involvement of healthcare consumers in the co-management of their own health is an essential component of sustainable health and healthcare and central to evidence-based practice [1]. However, effective participation is dependent on access to reliable information, and acquiring the knowledge, motivation, and competence to assess, understand, evaluate, and act upon the information available [2]. The World Health Organization described health literacy as the main desired outcome of health education, as an asset in itself, and as a public health issue [3].

There is growing evidence that health consumers increasingly rely on the internet for health information [4–7]. The internet offers a powerful means for obtaining information about medical conditions and their treatments [7]. A recent study by Morley et al. in 2020 has recognised the infosphere as a social determinant of health [3]. It was included in Dahlgren and Whitehead's model of health determinants [8]. Using this model, the determinants are the general socioeconomic, cultural, environmental, and now informational conditions [3].

Digital health literacy is the ability to use information and communication technologies to find, evaluate, create, and communicate health information, requiring both cognitive and technical skills [4]. Digital health literacy skills include computer and internet literacy, information literacy, functional literacy, and navigational and critical skills literacy [5]. Computer literacy refers to the ability to operate in computer-supported environments and includes both technical competence and an understanding of how information is presented electronically [9,10]. Information literacy is knowing when and why you need information, where to find it, and how to evaluate, use and communicate it [11,12].

Barriers to consumer use of online health information include lack of physical access to the computer and internet, motivation, and a real or perceived lack of knowledge, cognitive skills to find and use information [12–15]. Of these, motivation and skills are critical. Simply increasing access to online information will not lead to increased use unless consumers are motivated and able to make effective use of the internet and the information they find [16]. The expectation that online health information will be useful is one of the strongest predictors of internet use, and a lack of skills is debated as one possible reason for not using the internet.

Specifically, potential users may lack the skills to effectively locate, evaluate and use online health information depending on several factors [17–19].

Previous studies identified many critical factors in determining consumer's digital information-seeking behaviour, such as demographic characteristics (age, gender, educational status, socioeconomic class, and employment status) and health status [20]. Older adults are reluctant to use the internet, which is consistent with offline gender-based health-seeking behaviour [21]. Individuals who are better educated, employed, and of a high socioeconomic class prefer to use websites as a source of health information [22]. Those with poor/ fair health, chronic conditions, and long-term disability are more likely to visit health websites than those who reported better health [23]. A study targeting rural digital immigrants in China [24] reported that lack of health information literacy and low readability of digital health information are significant barriers for digital immigrants seeking and evaluating online health information.

Evaluating the trustworthiness of digital health information has been a critical issue in consumer health informatics research for many years [25]. A recent review done in Finland (2020) showed that participants face many challenges in evaluating the reliability of online health information [26]. Although they understand that online information should not always be trusted, they remain unsure of how to evaluate its credibility [27–29]. Another study by Metzgar et al. (2007) reported that internet consumers rely on either central or peripheral strategies for health information evaluation depending on high or low motivational status, respectively [30].

As such, many interventions are implemented targeting layperson's ability to identify reliable online information such as educational programs, interactive workshops, health literacy curricula, community outreach programs, and online portals [31]. All those interventions aim to teach people how to navigate online and then measure the change in knowledge and internet skills for the purpose of evaluation.

To our knowledge only two reviews have been conducted since 2010 to explore interventions targeting lay people. Car et al (2011) conducted a systematic review on interventions for enhancing consumers' online health literacy and found two interventions [16]. This was followed by Lee et al. comprehensive review (2014) on interventions to assist health consumers to find reliable online information which reported seven publications [31]. This review was unable to follow systematic review methods due to the paucity of research and humanistic interventions reported.

Based on the Information, Motivation and Behavioural Skills (IMB) model [16], we hypothesise that enhancing consumers' online health literacy has a positive effect on health behavioural outcomes. We aim to carry out a new systematic review of the topic given that the internet has changed a lot since the last review was carried out. To our knowledge, our systematic review paper is the first to explore the implemented interventions for lay people and their effects on e-health literacy skills and beliefs, and health outcomes.

## Aim and objectives

**Aim.**   To explore the useability and impact of interventions designed to improve laypersons' ability to identify trustworthy health-related online information.
  **Objectives.**

1. Outline the interventions which are designed to improve laypersons' ability to identify trustworthy online health-related information.

2. Identify the barriers and facilitators for using the interventions by laypersons.

3. Understand the interventions effects on layperson's knowledge, skills, attitudes, and confidence to identify trustworthy online health-related information.

4. Understand the interventions effects on layperson's health outcomes.

## Materials and methods

The protocol for this study has been published [32]. The age of the participants is widened to include all adults 18 years and older to explore the implemented interventions for all adult age groups. Due to time constraints, patients and the public are not involved in the research.

Ethics approval and consent for participation are not required to conduct this systematic review paper. As this study is based on the analysis of previously published anonymous data and does not contain identifiable individual's personal data, consent to participate is not applicable.

This systematic review is reported based on the PRISMA (Preferred Reporting Items for Systematic Reviews and Meta-Analysis) checklist [33].

### Review questions

PICO (population, intervention, comparator, outcome) framework is used to build our research question [34].

**Population:** Adult laypersons (patients and their carers' and the public).

**Intervention:** Interventions to improve layperson's ability to identify trustworthy online health information.

**Comparator:** Studies with or without comparative groups are considered for this study.

**Primary outcome:** Understand the interventions effects on laypersons' Knowledge, skills, attitudes, and beliefs to identify trustworthy digital health information.

**Secondary outcomes**: Understand the interventions effects on laypersons' health outcomes.

This systematic integrative review addressed the questions below:

1. What are the types of interventions to improve laypersons' ability to identify trustworthy digital health information?

2. What are the barriers and facilitators for using the interventions by laypeople?

3. What are the interventions effects on layperson's Knowledge, skills, attitudes, and beliefs to identify trustworthy online health-related information?

4. What are the interventions effects on layperson's health outcomes?

### Study design

This integrative review includes peer-reviewed quantitative (observational and experimental), qualitative, and mixed-methods papers.

### Study setting

Any study reporting an intervention implemented among adult people globally has been considered for this review (Community-based, Institutional, any specific setting).

### Inclusion criteria

Table 1 presents the inclusion and exclusion criteria with the justification.

**Table 1. Inclusion and exclusion criteria with the justification.**

| Criterion | Inclusion | Exclusion | Justification |
|---|---|---|---|
| Sample | Human studies | Animal studies | Referring to patient and public involvement in health research. |
| Population | - Adult group of patients, carers, and the public.<br>- Studies targeting adult age group (at least 18 years of age) | Healthcare providers | Explore implemented interventions targeting different age groups of adult lay people (young and older adults). |
| Language | English, Spanish, and Portuguese | Other languages | To widen the scope of the findings. |
| Time period | Studies done between 2006–2023 | Studies done outside this time frame | Covering a wide range of literature for implemented interventions during this time frame (more than 90%) |
| Geographic location | International context | | Identify locations where interventions are implemented targeting lay people. |
| Study focus | -Articles that discuss implemented interventions for improving laypersons' ability to identify trustworthy online health information<br>- Articles that present information on health literacy before and after the intervention. | -Studies focusing on unimplemented programs such as policies, laws, and regulations of governmental and non-governmental organisations<br>-Articles targeting interventions for healthcare providers<br>- Articles that don't present information on health literacy before and after the intervention. | -Referring to the context of the interventions. Explore interventions targeting online information-seeking behaviour for lay people in any setting (Offline, Online).<br>-Identify gaps in using the implemented interventions by laypeople to help improve future interventions' design. |
| Type of article | Peer-reviewed journal articles quantitative (experimental and observational), qualitative, and mixed-methods studies. | Grey literature and website's publications | Time constraints and feasibility |

## Information sources

The search was conducted on the following electronic bibliographic databases: Ovid Medline, Embase, Cochrane database, Academic Search Complete, and APA psycinfo. In addition, a discussion with an additional research team member regarding exclusion and inclusion criteria at the outset of the systematic review process occurred. Then, the references of the included studies were searched manually to identify other eligible studies.

## Search strategy

A health science librarian helped to construct the search strategy. Medical Subject Headings (MESH) and free-text terms are applied. Truncation and adjacency searching were used to improve the sensitivity of the search, as appropriate.

The search was conducted on 06/05/2023 using different search terms to explore as many articles as possible that cover our review scope. The search was updated on 01/07/2024 and yielded no more articles fitting our criteria to include. S1 Table presents the search strategy and the different search terms used.

## Data management

Duplicates were removed using Endnote Reference Management Software (Clarivate), and additional duplicates not identified by the Endnote function were removed manually. The deduplicated data was then imported into Rayyan, a review software that allows multiple reviewers to work simultaneously and independently on study selection.

## Study selection

The eligible studies were identified in two stages: title and abstract screening and full-text review. Abstracts and titles were screened by two independent reviewers using the eligibility

criteria outlined above. Conflicts were resolved by consensus and adjudicated by a third independent reviewer if required. The full-text review of the studies selected during the screening was independently conducted by two reviewers, with disagreement resolved as described previously.

## Data extraction

One reviewer extracted the data from each eligible study using a Microsoft Excel data extraction sheet. The following study details were extracted: authors, year of publication, country, and study design. Data was extracted about the types of implemented interventions, the outcomes of the implemented interventions and whether they are effective, and the barriers and facilitators for using the interventions by consumers. Particular results of the intervention outcomes were not reported at all ('selective non-reporting of results') or were reported incompletely ('selective under-reporting of results, e.g., stating only that "P >0.05" rather than providing summary statistics or an effect estimate and measure of precision), introducing a selective (non-) reporting bias. Some authors have been contacted to provide complete results; no replies have been received.

## Quality assessment

The mixed methods appraisal tool (MMAT) (version 18) was adopted to integrate quantitative, qualitative, and mixed-methods evidence. This tool is comprised of two screening items, followed by 15 appraisal items in three sections, including (1) five items on the Randomised control trials, (2) five items on the quantitative descriptive component, (3) five items on the nonrandomised studies. Reviewers independently evaluated the risk of bias of all selected articles with the MMAT, on an item-by-item basis. There are three response options: 'Yes' meaning the criterion is met, 'No' meaning the criterion is not met, and 'Can't tell' when there is not enough information in the paper to judge whether the criterion is met. Responding 'No' or 'Can't tell' to one or both screening items indicates that the paper is not an empirical study, and thus couldn't be appraised using the MMAT. Since there are only a few items for each domain, the score was presented using descriptors (%): 100% quality criteria met, 80% quality criteria met, 60% quality criteria met, 40% quality criteria met, 20% quality criteria met. The quality of the paper is rated good, moderate, or poor when 100% of quality criteria are met, 80% or 60% of quality criteria are met, 40% or 20% of quality criteria are met, respectively [35].

The MMAT was used to assess the quality of studies with different methodological designs. Two reviewers independently appraised the quality of each included study. Disagreements between reviewers were resolved by discussion. Rating 'Can't tell' to the appraisal items has led the reviewers to look for companion papers and contact the authors for more information and clarification. If the information is complete to address the item, the reviewers agreed to rate the item as "Yes"; otherwise, we assigned a "no" response. However, none of the review authors is methodologically expert in all study designs, which may have introduced a degree of bias.

## Data synthesis

Due to the different methodological designs of the included studies, the findings of the narrative synthesis were provided from the included studies, focused around answering the main review questions. We presented the summarised information on studies reporting implemented interventions aimed at improving users' ability to identify trustworthy online health information according to preidentified thematic areas such as the study settings, target

population, intervention type, style and content, intervention theory, outcomes of the implemented intervention, and the barriers and facilitators for using the interventions by consumers. Furthermore, Whittemore and Knafl's integrative review approach was used as guidance for narrative synthesis [36]. The intervention's outcomes were summarised according to the reported change in the e-health literacy knowledge, skills, attitudes, and beliefs and the health outcomes after the intervention (mean difference) and statistically significant/insignificant results (P value). Furthermore, a narrative synthesis of the barriers and facilitators for using the implemented interventions was provided.

## Results

The total records identified from all databases were fourteen thousand two hundred and sixty-five. Three thousand and forty-four records were removed before screening. Of the eleven thousand two hundred and twenty-one records screened, eleven thousand and one hundred fifty-six were excluded. Reports sought for retrieval and then assessed for eligibility were sixty-five. Reports excluded were forty-three. The final studies included were twelve (Fig 1).

Most of the studies (n = 7) 58.4% included < 100 participants [2,9,37–41]. Few studies 3 (25%) has participants between 100–1000 [10,42,43]. Only one study (n = 1) 8.3% has a large sample size, >1000 [44] (S2 Table). Most of the studies (n = 8) 66.6% included young adults [2,10,37–39,42–44]. Three studies (n = 3) 25% included old adults [9,40,45] and only one study (n = 1) 8.3% included both age groups [41] (S3 Table). Randomised control trial is the predominant study design (n = 7) 58.4% [2,10,37,39,40,43,44], followed by the quantitative design (n = 4) 33.3% [9,38,41,45], and only one study (n = 1) 8.3% used a quasi-experimental

**Results:**

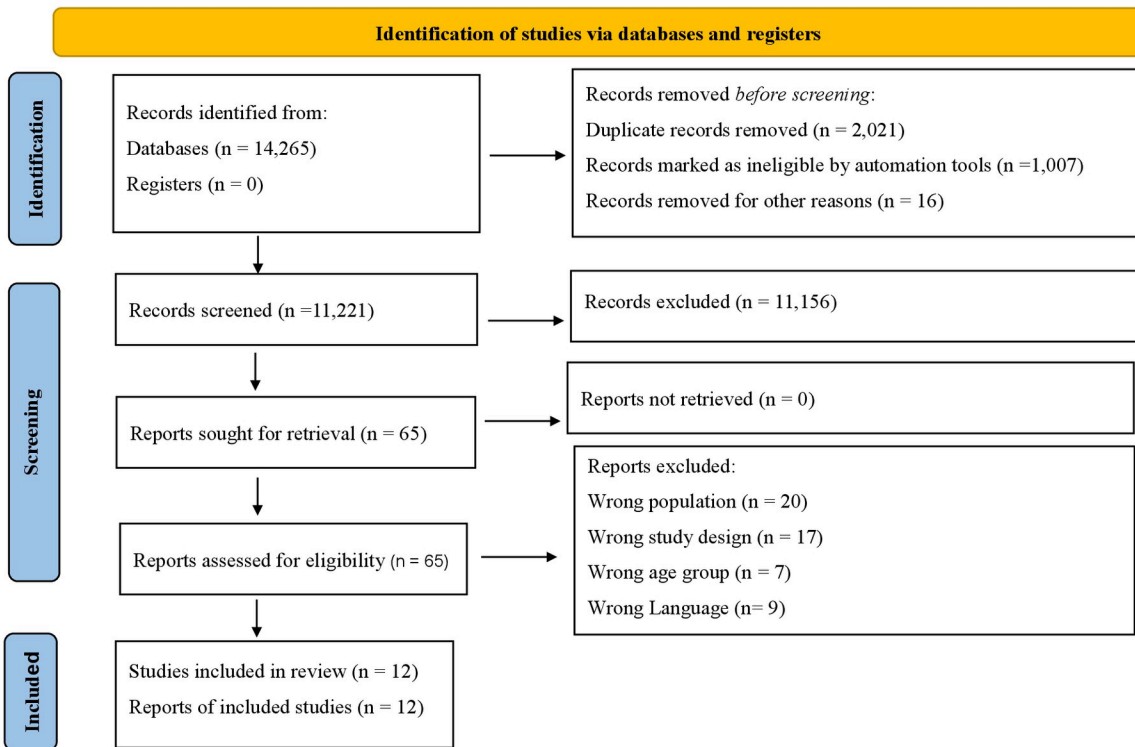

**Fig 1. PRISMA 2020 flow diagram for new systematic reviews including searches of databases and registers only.**

design [42] (S4 Table). Half of the studies presented implemented interventions in USA
(n = 6) 50% [40–45] only one study (n = 1) 8.3% was done in each of the following countries:
Germany [37], Australia [38], Norway [2], Georgia [10], Canada [9], 2006, and UK [39] (S5
Table). The number of the studies on implemented interventions was increasing every five
years, n = 2 (17%) (2006–2011) [9,10], n = 4 (33%) (2012–2017) [2,37,38,45], and n = 6 (50%)
(2018–2023) [39–44] (S6 Table).

As seen in Table 2, media literacy interventions constitute the majority n = 8 (66.6%)
[2,9,10,37,38,40,42,45]. Other intervention types are boosting scientific reasoning [44], accu-
racy Nudging [43], enriching Wikipedia content [39], and pre-activation prior topic knowl-
edge tool [41], each constitute n = 1 (8,3%). The commonest type of media literacy
interventions is the web-based training sessions n = 3 (25%), followed by interactive work-
shops [9,38] and multimedia education [40,42], each constitutes n = 2 (16,7%), and the minor-
ity is the outreach educational program n = 1 (8.3%) [45]. Other intervention types are top-
down intervention [39], boosting intervention [44], accuracy nudging intervention [43], and
pre-activation tool [41]. The intervention style varies across the studies, n = 2 (16.7%) inter-
ventions are internet-skills building [9,10] and 2 (16.7%) are competence-based [44]. Other
intervention styles include group sessions training [38], source evaluation training [37], an
interactive online skill-based learning (web portal) [2], an educational learning series [45], e
health tutorial [40], an educational video [42], enriching Wikipedia content [39], and prob-
lem-based search learning [41], each constitute 1 (8.3%). There are different methods used for
health information evaluation. N = 4 (33.3%) interventions used specific criteria such as the
Health on the Net Foundation [9], SEEK (Source, Evidence, Explanation, and Knowledge)
[37], the full DISCERN [2], and the PILOT (purpose–information–links–originator–timeli-
ness) criteria [10]. Participants are instructed on how to use MedlinePlus to find health infor-
mation in two interventions (16.7%) [40,45]. 16.7% of the interventions (n = 2) have no
information about the method used for evaluation [38,42]. Other methods used for health
information evaluation are self-rated accuracy of a single headline [43], infographic guided
accuracy [44], posting the relevant Cochrane review's summary tables with references on the
Wikipedia page [39], and search by generating keywords [41], each constitutes 1 (8.3%). There
is variation across the studies for the theories/models used to reflect how the intervention
works (n = 8) 66.7%, examples (an integrated model of epistemic beliefs [37], (conceptual
framework of shared decision-making and evidence-based practice model/ theory of planned
behaviour/ multi-dimensional model) [2], social–cognitive theory [10], the cognitive theory of
multimedia learning [40], gateway belief model [44], inattention-based account of misinfor-
mation sharing on social media [43], Orem's theory of self-care [42], model of information
search with a search engine [41]. Few studies didn't report a theory (n = 4) 33.3% [9,38,39,45].
Most of the interventions (n = 8) 66.7% are targeting the general public {(young adults (n = 4)
[2,37,43,44], older adults (n = 2)[9,40], young and older adults (n = 1) [41], no information
(n = 1) [38]}, (n = 3) 25% are designed for the patients {HIV young patients (n = 2) [10,42],
patients with schizophrenia and related psychotic conditions (n = 1) [39]}, and the minority
(n = 1) 8.3% is targeting both the patients and public old adults [45]. S7 Table shows a detailed
summary of intervention's description.

A total of two authors independently appraised the twelve articles that met the inclusion cri-
teria for methodological quality. The S1 Checklist summarises the questions answered in the
mixed method appraisal tool checklist (version 18) [46]. Of the twelve included studies, only
two studies (16.7%) scored well on the quality assessment (100% of the quality criteria met),
including one quantitative study [38] and a quasi-experimental (non-randomised control
trial) design [42]. The rest of the studies 10/12 (83.3%) scored moderate, with 60% and 80% of
the quality criteria met, including all the randomised control trials (7/12) [2,10,37,39,40,43,44]

**Table 2.** Intervention types, styles, method used for health information search and/or evaluation, theories, and target audience.

| Intervention type | | No. of studies (%) | Intervention styles | Method used for health information search and/or evaluation | Intervention theory | Target audience |
|---|---|---|---|---|---|---|
| **Media literacy interventions** | Interactive workshops | 2 | Group sessions training [38] | No information | No information | General public |
| | | | Internet skills building–intervention [9] | Participants were introduced to health and cancer web sites of various top-level domains (.gov,.com,.org,.edu) and Web pages accredited by the Health on the Net Foundation, such as the National Cancer Institute and the American Cancer Society | No information | General public (older adults) |
| | Web-based training | 3 | Short source evaluation training [37] | As in the SEEK intervention, The criteria used for evaluation are (Source, Evidence, Explanation, and Knowledge) / Comparison of information across different information sources. | An integrated model of epistemic beliefs. | General public (young adults) |
| | | | Interactive online skill-based learning (web portal) [2] | Participants were asked to rate the trustworthiness of an online article using the full DISCERN critical appraisal tool which was provided on the web portal. | Conceptual framework of shared decision-making and evidence-based practice model/ Theory of Planned Behaviour/ multi-dimensional model | General public (young adults) |
| | | | Internet skills-building intervention (behavioural intervention) [10] | Participants were instructed in criteria for evaluating the quality of information obtained online (the PILOT criteria). | social–cognitive theory. | HIV patients (young adults) |
| | Community outreach education [45] | 1 | An educational learning series | Participants were instructed on how to use MedlinePlus to find health/drugs/supplements information. | No information | Patients and general public (older adults) |
| | Multimedia education | 2 | E health tutorial [40] | Introduction to the MedlinePlus.gov website, use of the health Topics and the Drugs and Supplements sections on MedlinePlus.gov. | The cognitive theory of multimedia learning | General public (older adults) |
| | | | Educational video [42] | No information | Orem's theory of self-care | HIV patients (young adults) |
| Top-down intervention [39] | | 1 (8.3) | Enriching Wikipedia content | Posting the relevant Cochrane review's Summary of Findings table on the target Wikipedia page along with references to the review's web page and full text. | No information | Patients with schizophrenia and related psychotic conditions. |
| Boosting intervention [44] | | 1 (8.3) | Competence-based intervention | Participants were exposed to 10 statements, of which five were scientifically accurate and five were at odds with the best available evidence. Participants responded by indicating the accuracy of a statement using an infographic. | Gateway belief model | General public (young adults) |
| Accuracy Nudging intervention [43] | | 1 (8.3) | Competence-based intervention | Participants self-rated the accuracy of a single headline before beginning the news-sharing task. Each participant saw one of four possible headlines. | Inattention-based account of misinformation sharing on social media | General public (young adults) |
| Pre-activation tool [41] | | 1 (8.3) | Problem-based search learning | Participants started by reading the search problem statements, then they were instructed to produce three keywords related to a concept extracted from the search problem and to the search problem statement, then they were instructed to imagine three different keywords that might be useful to search for the answer. | Model of information search with a search engine | General public (young and old adults) |

and the remaining three quantitative study designs [9,41,45]. All the randomised control trials were rated of moderate quality as they didn't report if the outcome assessor is blinded to the intervention provided [2,10,37,39,40,43,44], and half of them didn't mention if the randomisation is appropriately performed [37,34,44], which could introduce performance and selection bias, respectively, and affect the synthesis and generalisability of our findings. We contacted authors for the missing information, but no replies were received.

As in Table 3, quarter of the studies (n = 3) 25% has procedural and/or navigation/evaluation skills building style [9,10,40], while the majority (n = 7) 58.3% has navigation/evaluation skills building style [2,37–39,41,42,45], and (n = 2) 16.7% is competence- based [43,44]. Different outcomes are measured across the studies, including the knowledge, skills, attitude, and beliefs about evidence-based health information, and the health outcomes. 25% (n = 3) of the studies measured the knowledge including computer/internet knowledge [40], internet knowledge [9], knowledge about evidence-based health information [38], each constitutes (n = 1) 8.3% and all reflects a statistically significant result.

Most of the studies 83% (n = 10) measured the skills including the evaluation skills (n = 5) 41.7% [2,10,37,40,41], navigational skills (n = 4) 33.3% [38,39,43,45], e health literacy self-efficacy (n = 3) 25% [9,10,40], and the procedural skills (n = 2) 16.7% [10,40]. All the studies measuring the procedural and e health literacy efficacy reflects a statistically significant result. Most of the studies measuring the evaluation skills has statistically significant results (n = 4) 80% [10,37,40,41], only one study (n = 1) 20% has non-significant result [2]. Half (n = 2) 50% of the studies measuring the navigational skills has statistically significant results) [38,43],

**Table 3. Intervention outcomes.**

| Intervention style/ % | Intervention type/reference | Intervention outcomes | | | | |
|---|---|---|---|---|---|---|
| | | Knowledge | skills | Attitudes | Beliefs | Health outcomes |
| (Procedural and/or navigation/ evaluation skills building intervention (25%) | Behavioural intervention [10] | NA | -**Procedural skills:** The mean value for internet use yielded a statistically significant increase at 3, 6, and 9 month follow ups. -**E-health literacy self-efficiency:** A statistically significant increase in the mean value of self-efficiency for health information use at 3, 6, and 9 months follow ups. -**Health information evaluation skills:** A statistically significant increase in discrimination between the two presented web pages at the 3-month follow-up was observed. | NA | NA | -A statistically significant increase in **social support** mean at 3,6,9 months. - A statistically significant reduction of the mean value of **affective depression** at 6 month follow up |
| | Public library workshop [9] | -A statistically significant increase in standard error **of internet Knowledge** after 4 months (p < 0.001). | -**E health literacy self-efficacy:** Participants had higher self-efficacy with internet searching skills. A statistically significant increase in standard error of ease of internet searching after 4 months (p < 0.01). | **Satisfaction with the intervention:** High satisfaction with the workshop format and content was reported after 4 months. **E health literacy confidence:** Increase in participants confidence to search online independently was noted after 4 months. | **E health literacy beliefs:** 70% of the participants indicated that they would rely on the Internet for cancer information in the future (no more information about unit of measurement) | NA |
| | e health tutorial [40] | -**A statistically significant increase in computer/internet knowledge** after 2 weeks (p < .001). | -**Procedural skills:** A statistically significant increase in computer/ internet use mean score after 2 weeks (p < .001). -**A statistically significant increase in eHealth literacy efficacy** mean score after 2 weeks (p < .001). -**A statistically significant increase in Health information evaluation skills** mean score after 2 weeks (p < .001). | -**Satisfaction with the intervention:** The intervention showed a positive attitude towards training | NA | NA |

(*Continued*)

**Table 3.** (Continued)

| Intervention style/ % | Intervention type/reference | Intervention outcomes | | | | |
|---|---|---|---|---|---|---|
| | | Knowledge | skills | Attitudes | Beliefs | Health outcomes |
| (Navigation/ Evaluation) skills building interventions | Source evaluation training intervention [37] | NA | - **Evaluation skills:** The intervention group spent a statistically significant more time on objective web pages than the control group (p = .002). Furthermore, they spent a statistically significant less time on subjective and commercial web pages than controls (p < .001).). | -**Post search decision confidence:** Participants in the intervention group have developed a statistically significant more confidence of their decision than controls (p = .018). | - **Internet specific epistemic beliefs:** 1 week after the intervention participants possessed statistically significant stronger beliefs that Internet-based knowledge claims need to be critically evaluated through cross-checking (p = .003). | NA |
| | Pharmacy Community Outreach Program [45] | NA | **Navigation skills:** Participants indicated they were very likely to use MedlinePlus to find health information (83%) and to find information on drugs and supplements (79%) after the intervention. | **E health literacy Confidence:** 65% strongly agreed they were more confident in their ability to evaluate the reliability of online health information after the intervention. | NA | Improved medication adherence leading to reduction of participant's haemoglobin A1c and blood pressure |
| | Community educational initiatives [38] | -A statistically significant improvement in participant's knowledge **about evidence-based health information** (p = 0.023). | -**Navigation skills:** A statistically significant improvement in participant's **skills with finding and using evidence-based health information.** | Most participants strongly agreed or agreed (78.2%) that the intervention improved **their attitude towards searching evidence-based health information.** | **E health literacy beliefs:** 52.8% strongly agreed or 40.5% agreed that the intervention would **change the way they looked for and used health information in future (p = 0.051).** | NA |
| | **Web portal** [2] | NA | **Evaluation skills:** Only minor improvement in the intervention group was identified and this difference was not statistically significant (p = 0.90) after 3 weeks. | -A statistically significant difference was found for overall **attitude towards search** in favour of the intervention group (p = 0.03) after 3 weeks. -**Satisfaction with the intervention** was good | **E health literacy beliefs:** small statistically non-significant improvement after 3 weeks in favour of the intervention group was noted indicating higher intention to search and higher activation. | NA |
| | **Educational video** [42] | NA | NA | NA | NA | -No statistically significant differences in self-care agency after the intervention. -No change in SCI (Self-as-carer-inventory) scores over the 1-week period. |
| | **Enriching Wikipedia content** [39] | NA | **Navigation skills:** The point estimates for the ratio of geometric means favoured the intervention group for the number of visits to the free summary page and the number of full-text downloads) but no statistically significant evidence of an effect (P value = 0.39, 0.69 respectively) at 12 month. | **Attitude towards search:** Altmetric score (a measure of social media activity) indicated some evidence of an intervention effect (statistically significant: p value 0.02) at 12 month. | NA | NA |
| | **Prior topic knowledge pre-activation support tool** [41] | NA | **Evaluation skills:** Participants who completed the semantic pre-activation before searching the internet spent longer time evaluating the search engine results page (M = 148.13 SD = 16.12) than the control (M = 100.72 SD = 16.90). Effects of semantic pre-activation was statistically significant (P <0.05) after 3 weeks. | NA | NA | NA |
| Competence building intervention | Boosting consensus reasoning [44] | NA | NA | NA | -**Belief accuracy:** The boosting intervention yielded no statistically significant increase in the mean belief accuracy score over time (1,2,3 weeks). | -Accurate beliefs were statistically significant correlated with self-reported behaviour aimed at preventing the coronavirus from spreading (P value = 0.001) |
| | Accuracy nudging intervention [43] | NA | **Navigational skills:** Sharing intentions for true headlines were statistically significant higher than for false headlines in the intervention group after 3 days (p < .0001). The intervention increased sharing discernment. | NA | **Belief accuracy:** 3 days after the intervention the accuracy nudge makes participants more likely to consider accuracy when deciding whether to share. | NA |

*Intervention effectiveness

-**Green highlight**: Effective interventions (all statistically significant outcome results).

-**Yellow highlight**: Partially effective intervention [no statistical information (p value)] OR partially effective (statistically significant and non-statistical outcome results).

-**Red highlight**: Non-effective interventions (Statistically non-significant).

(n = 1) 25% has non-significant results [39], and (n = 1) 25% was effective but no statistical information [45].

58.3% (N = 7) of the studies measures the attitudes, including satisfaction with the intervention [2,9,40], e-health literacy/post search confidence [9,37,45], and attitudes towards search [2,38,39], each constitute (n = 3) 25%. Satisfaction with the training is positive in the three studies. E- health literacy/post search confidence is improved in the three studies, one of them has a statistically significant result [37]. The attitude towards search is improved in the three studies with two studies reflecting statistically significant results [2,39]. Most of the studies (n = 5) 71.4% measuring the e-health literacy attitudes are significantly effective.

Half of the studies (N = 6) measures the beliefs including e-health literacy beliefs (n = 3) 25% [2,9,38], epistemic beliefs (n = 1) 8.3% [37], and belief accuracy (n = 2) 16.7% [43,44]. E-health literacy beliefs are improved in the three studies, although one is statistically non-significant [44]. Internet specific epistemic beliefs are improved significantly. The belief accuracy is improved in both studies although one is statistically non-significant [44]. Most of the studies (n = 4) measuring beliefs in using the internet for health literacy are significantly effective.

Third of the studies (n = 4) 33.3% measures the health outcomes including (social support/ affective depression) [10], haemoglobin A1C and blood pressure [45], self-care agency [42], and corona virus prevention [44], each constitutes (n = 1) 8.3%. A significant increase in social support mean was reported at 3,6,9 months with statistically significant reduction of the mean value of affective depression at 6 months follow up. Improved medication adherence was reported leading to reduction of participant's haemoglobin A1c and blood pressure (no statistical information). No change in self-care agency was seen one week after the intervention (non-significant). Accurate beliefs were correlated with self-reported behaviour aimed at preventing the coronavirus from spreading (P value = 0.001). Few studies measure health outcomes [10,42,44,45], the minority (n = 1) 25% are significantly effective [10].

All the Procedural and/or navigation/ evaluation skills building interventions (n = 3) 25% are effective with statistically significant results [9,10,40]. Half (n = 1) 50% of the competence-based interventions is partially effective (significant + non-significant results) [44], while the other half is effective [43]. 42.8% (n = 3) Half of the navigation/ evaluation skills building interventions is effective [37,38,41] and (n = 3) 42.8% the other half is partially effective [2,39,45]. Only one study 14.3% has non-significant change of the results after the intervention (non-effective) [42]. S8 Table presents detailed description of intervention's measures, outcomes, effectiveness, and critique.

As seen in Table 4 many barriers and facilitators for intervention use by consumers are reported across the studies, which are related to participants characteristics, incentives, and intervention setting, delivery, design, material, and duration.

Basic Internet skills and experience on using Facebook and Twitter are reported as barriers for participation in 25% (n = 3) of the interventions [37,41,43], while 16.7% (n = 2) of the interventions didn't require that, as they involve internet skills building training [9,10]. Low participant's Health/reading Literacy is a barrier in (16.7%) (n = 2) of the interventions [2,10], while high educational status facilitates intervention use in n = 3 (25%) of the studies [40,41,44].

Participants are given incentives either in the form of payment or a training certificate in (n = 5) 41.7% of the interventions, this facilitated participant's commitment to the intervention during the start and follow up waves [10,37,41,44,45]. Concerning the intervention setting, three interventions (25%) are implemented as experiments in a lab context, this limited their public accessibility [40,41,43]. Lacking cultural diversity is an obstacle in (n = 5) 41.7% of the studies as they are conducted in English with an English-speaking population [9,10,40–42].

**Table 4. Barriers and facilitators for intervention use by consumers.**

| | Barriers | Number of studies + references | Facilitators | Number of studies + references |
|---|---|---|---|---|
| **Participant's digital skills level** | - Basic computer/internet skills/experience on using Facebook and Twitter were required for participation in web-based interventions | **3 (25%)** [37,41,43] | - Computer experience was not required for participation. | **2 (16.7%)** [9,10] |
| | - Low health/reading literacy/ | **2 (16.7%)** [2,10] | - High educational status | **3 (25%)** [40,41,44] |
| **Incentives** | | | -Participants were given incentives (payment or certificate). | **5 (41.7)** [10,37,41,44,45] |
| **Intervention setting** | -The intervention was implemented as an experiment in a Lab-context (not publicly accessible). | **3 (25%)** [40,41,43] | -Intervention held at settings tailored to the participants (senior centres, CTAC, public library). | **4 (33.3)** [9,38] |
| | -The intervention is conducted in English with an English-speaking population, lacking cultural diversity | **5 (41.7%)** [9,10,40–42]. | - Real-life setting. | **8 (66.6)** [2,9,10,37,38,40,44,45] |
| **Intervention style** | - Complex problem-based search learning that required users to engage cognitive resources to make inferences, assess information relevance and select websites to visit. | **1 (8.3%)** [47] | Interactive small group learning sessions. | **4 (33.3%)** [9,10,38,40] |
| | | | Competence -based intervention. | **2 (16.7%)** [43,44] |
| **Intervention delivery** | - Exclusively web-based intervention, not guided by facilitator. | **5 (41.7%)** [2,43,45,47,49] | -Intervention delivered free through collaboration and involved consumer health Liberian and/or healthcare providers (pharmacists, nurses). | **4 (33.3%)** [9,38,42,45] |
| | | | Intervention is guided by a facilitator. | **4 (33.3%)** [9,10,40,45] |
| **Intervention design** | - The intervention design is not appropriate (presented in a tabular format/ Ill-defined search problems). (presented in a tabular format [39]/ ill-defined search problems [41] | **2 (16.7%)** [39] | -The intervention is designed from patients and public perspectives/ with experts' consultation/ human-centred. | **3 (25%)** [40] |
| | | | -The intervention framework was designed as pre-test. | **1 (8.3%)** [43] |
| **Intervention material** | -The topic is not of interest to the participants/ Information overload/ priori Complex material. | **4 (33.3%)** [9,38,39,41] | The topic used for the evaluation task is of interest to the participants/Material is relevant/understandable/ readable (plain English). | **6 (50%)** [2,9,10,38,39,43–45] |
| | | | Material was sourced from evidenced-based reliable resources. | **9 (75%)** [2,9,10,39,40,42–45] |
| | | | Guidance/instructions was provided on how to use the intervention/ evaluate the claim. | **8 (66.6%)** [2,9,10,40,41,43–45] |
| **Intervention duration** | -Inappropriate intervention duration (too Short /too long duration to access the intervention)/ interval between sessions. | **6 (50%)** [2,37,40,42] | - Appropriate intervention duration/interval (Self-paced/ Unlimited by time/ | **3 (25%)** [9,41,42] |

The majority (n = 8) 66.6% of the interventions are implemented in a suitable real life setting [2,9,10,37,38,40,44,45] and half of them are held at a setting tailored to the participants [public libraries (n = 2) 50% [9,38], senior centres (n = 1) 25% [45], a CTAC (Community technology access centre) at an AIDS service agency (n = 1) 25% [10]. Intervention styles vary across the studies, (n = 4) 33.3% of the interventions are presented in an interactive small group context to enable a personalized and relaxed training environment [9,10,38,40] and (n = 2) 16.7% used a unique competence-based approach [43,44]. Only one intervention is presented as a complex problem-based search learning that required users to engage cognitive resources to make inferences, assess information relevance and select websites to visit [41]. Almost half of the interventions (n = 5) are delivered as an exclusive web-based intervention, not guided by facilitator, this limited their use [2,37,39,41,43]. In contrast third of the

implemented interventions are guided by a facilitator [9,10,40,45]. 33.3% (n = 4) are distinguished interventions which are delivered through collaboration of multiple stakeholders in health [9,38,42,45], half of them (n = 2) 50% are conducted by healthcare providers [pharmacists [45], nurses [42]. Third of the interventions (n = 4) 33.3% has a unique design, using a participatory design user-centred approach [40], involving patients and public [2], expert consultations [39], and has a framework designed as pre-test [43]. Two interventions 16.7% have inappropriate design (presented in a tabular format [39]/ ill-defined search problems [41]. The topic used for the evaluation task is of interest to the participants in the majority of the interventions (n = 8) 66.6% [2,9,10,38,39,43–45]. The topics used are swine flu vaccine (n = 1) [2], HIV management (n = 2) [10,42], COVID 19 (n = 2) [43,44], cancer prevention (n = 1) [9], and dealing with multiple medications (n = 1) [45]. In contrast, material is reported to be complex and not of interest to the participants in third of the interventions (n = 4) 33.3% [9,38,39,41]. The intervention materials are sourced from evidenced-based reliable sources in most of the interventions (n = 9) 75% [2,9,10,39,40,42–45], almost (n = 8) 88.8% of them has a guidance/ instruction provided on how to use the intervention and/or evaluate the claim [2,9,10,40,41,43–45]. Inappropriate intervention duration/ interval between sessions has limited the intervention use in n = 6 (50%) of the studies [2,10,37,38,40,42], while quarter of the studies are facilitated by appropriate intervention duration [9,41,42]. S9 Table shows detailed description of participant's characteristics and the barriers and facilitators for intervention use by consumers.

Media literacy interventions are the most common type of the implemented interventions. Few studies measure the effect of the intervention on health outcomes, the minority is significantly effective. All the procedural and navigation/ evaluation skills building interventions are effective with statistically significant results. The main barriers for layperson's participation in web-based interventions are lack of participant's computer/internet experience and absence of guidance/facilitator. Few interventions are delivered in a suitable location for the participant's, in addition the minority involved lay people in designing the intervention.

## Discussion

### Summary of main results

We found media literacy interventions dominated in being the most implemented intervention type. Few studies measured the effect of the interventions on participant's health outcomes, the minority are effective. All the procedural and navigation/ evaluation skills building interventions are significantly effective. Lack of participant's computer/internet experience and absence of guidance/facilitator are major barriers for participating in web-based interventions. A gap was noticed in providing the suitable location of delivery of the educational intervention, in addition to using human-centred approach for designing the interventions.

### Agreements and disagreements with other studies

**Intervention types.**   The dominance of media literacy intervention was reported in studies done by Car et al. and Lee et al. in UK and Australia (2011, 2014), respectively [16,31]. Both publications explored exclusively media literacy interventions. However, the trend of interventions has changed during the last five years (2018–2023) focusing on competence-based, top-down, and problem-based search learning approaches. Our paper is the first to explore those types. However, more evidence is needed to determine the effectiveness and scalability of such interventions.

**Interventions effects on health outcomes.**   We also noted few studies measuring the impact of digital health literacy interventions on the patients' health outcomes [10,42,44,45];

the minority are effective [10]. This is supported by a study done recently in Australia (2021) on the effects of web-based portals on parental health knowledge and child health outcomes [47]. This disagrees with Car et al.'s (2011) study findings [16], as the two included studies measured the participants' health outcomes, and half of them were statistically effective. Our review reported the implemented interventions since 2006 and conversely explored twelve studies, third have measured the health outcomes, however, only one intervention is statistically effective. In addition, it is hard to compare across the studies due to differing baseline health literacy in different context.

**Intervention style effectiveness.** Our review found the procedural and navigation/ evaluation skills-building interventions are effective with statistically significant results and predominantly targeting older adults. This finding is also seen in a study done by Xie et al. (2011) which reflects the statistically significant effect of the interactive workshops on participants computer/web knowledge and skills and e health literacy [48]. This agrees with the results of Xie et al. (2012) study that reveals the positive impact of computer training on older adult's e-health literacy [49].

**Barriers to intervention use.** Limited computer/internet skills are reported as a barrier for intervention use in most of the web-based interventions. This is supported by Kelley et al study done in Canada (2007) that greater exposure and prior experience of computers and the internet are reported as significant predictors of search performance [50].

Absence of a facilitator (consumer health Liberian/healthcare providers) is also a common barrier for intervention use. A trained facilitator can direct the participant and provide guidance as needed. This agrees with the findings of Parker et al (2005) and Voigt-Barbarowicz et al (2020) that emphasise the role of consumer health librarian and healthcare providers in health literacy education [51,52].

Addressing the socioeconomic disadvantage needs considering implementation of digital health interventions was not considered in our paper. This has been highlighted in Azzopardi-Muscat study in 2019 [53] that emphasize the importance of implementing programmes to enhance digital health literacy as well as monitoring utilisation and impact across all groups in the society to reduce health inequalities. This was also considered in a study done in 2020 by Kemp et al. [54] that recommend system-based approaches to enhance digital inclusion. Future research should use the health literacy lens to understand the interaction between personal, contextual, and technological factors to determine the consequent use of digital technologies for health.

**Facilitators of intervention use.** The minority of the studies used a unique human-centred approach in the design process of the intervention. This finding was noted in a study published in 2007 by Gross et al that involves neurologist, instructional designer, and librarian in the design of the blended educational program [55]. This is consistent with Kurtz-Rossi and Duduay study as it involves health literacy experts in the design of school health literacy curriculum intervention [56].

The end users should be involved in the design process of future educational interventions to ensure participant's motivation. This is consistent with our proposed information, motivation, and behavioural (IMB) model [57]. The IMB model assumes that information and motivation influence behavioural skills, which subsequently influence health behaviours. Most studies suggest that information on its own has limited direct health benefits or ability to change behaviour [58,59]. Following Kalichman's hypothesised route of action (Kalichman 2002; Kalichman 2006) [10,60] we suggest that interventions directly enhance internet skills, internet-related self-efficacy, and internet use, leading to increased health knowledge, a more active coping style, and improved information handling, which then may lead to improved health behaviours. This hypothesis aligns with research demonstrating that participation in

education and using the internet for health information are associated with positive health and social outcomes [61].

Few interventions are implemented in a context tailored to participants needs (public libraries and senior centres settings). This is consistent with two grey literature reports, ARCH [Schneider E. ARCH Evaluation Focus Groups. Boston (USA): Massachusetts General Hospital; 2009] and Medline in the mountains [Carlson G. NN/LM Quarterly Report. Colorado (USA): Poudre Valley Health System; 2003], where trainings were held at senior centre in Revere and Massachusetts and Patrons of Estes Park, Red Feather Lakes, and Wellington Libraries to ensure community engagement. Older adults have positive view of the public library and believe that libraries play an important role as health information providers. Senior centres provide opportunities of social networking and activities for older adults as well as health-related social support groups to gather.

## Potential biases

The MMAT is used to assess studies with different methodological designs. None of the quality assessment reviewers are experts in all types of study designs. Most of the included studies have a randomised control trial study design; all were rated of moderate quality as they didn't report blinding of the outcome assessor to the intervention and if the randomisation is appropriately performed. This could introduce performance and selection bias, affecting the synthesis and generalisability of our findings. Thus, there is a need for well-designed and rigorously conducted randomised controlled trials to understand the effectiveness of the interventions for improving digital health literacy. As our review includes complex interventions, outcomes are likely affected by participant's characteristics such as age, health status, and aspects of the intervention (for example, mode of delivery or setting).

## Implications for practice

Educational strategies to support people in improving their digital health literacy, focusing on addressing computer and internet illiteracy, involving facilitators for intervention delivery, and focusing on competence building and design thinking approaches, could have promising outcomes. Providing access to the computer/ internet without training is not sufficient to improve digital health literacy. This has implications for wider policy and suggests that investment in infrastructure needs to be accompanied by investment in training. Many potential agents must join forces to build the health literacy of the lay public- healthcare professionals, health service organizations, community centres, public libraries, and continuing education providers.

## Strength and limitations

We present different theories/models on how the interventions work [2,10,37,40–44]. This study includes adults of different age groups (young and older adults). Two studies implemented in USA used a recruitment platform that is reasonably representative while being affordable for large samples [43,44].

Per the protocol, we plan to search comprehensively and include studies in different languages. However, upon selection, neither Spanish nor Portuguese studies were found. As such, we included only English studies; this could lead to language bias. The included studies reported interventions implemented in different country settings; this may introduce a location bias and affect the generalisability of our findings.

The MMAT is a useful critical appraisal tool since it provides, within a single tool, methodological quality criteria for different study designs. Also, the MMAT focuses on a limited

number of core criteria, enabling a more efficient quality appraisal. The criteria in the MMAT are more difficult to judge than in other appraisal tools because they focus on methodological quality and not on reporting quality. Methodological quality criteria are more difficult to interpret because the reviewers need to judge whether the reported results of the study are trustworthy [35].

This review includes twelve studies, the majority had less than 100 participants [2,9,37–41] and targeting young adults [2,10,37–39,42–44]. The included studies tested interventions that were not comparable regarding content and format. Nonetheless, the quality of evidence must be downgraded by the fact that only twelve studies could be included, the majority had a small sample size. The interventions were implemented in different country settings, using different methodological designs, this could impact generalizability of our findings. Most of the studies had a Randomised control trial study design [2,10,37,39,40,43,44]. However, not all the outcomes are reported across the studies, in addition different instruments are used to measure reported outcomes, as such meta-analysis can't be conducted.

To our knowledge this is the first integrative systematic review study to explore the barriers and facilitators for using the e health literacy interventions by lay audience using the mixed method appraisal tool. We explore different methods used for health information evaluation such as Hone code [9], SEEK criteria [37], DISCERN [2] and PILOT criteria [10]. Our study went beyond exploring the knowledge, skills, attitudes, and beliefs of the participants to discover the impact of the interventions on the participants health outcomes.

## Conclusion

Few interventions reported using competence building approaches to teach people how to navigate online information. Most of the included studies focused on measuring digital health literacy skills and attitudes rather than the long-term effect of the intervention on participant's health outcomes. Only a handful of studies demonstrated improved health. One important finding of our study is that limited computer/internet skills are still a major barrier to using web-based interventions. We found a lack of descriptions of if and how end users were involved in intervention design. Few studies are noted to use design thinking approaches to facilitate intervention use by consumers. Future work could tailor interventions and context to participants needs. Designing future educational strategies with involvement of end users could play an essential facilitating role. There is considerable scope for further research to understand who is best involved in the design process. Implementing a practical educational intervention in a location convenient to the participants could facilitate intervention use by consumers. However, more evidence is needed to understand where the best place is to do these classes and when is the best time. Program facilitators could play a critical role in guiding laypeople how to navigate online health information. However, more research is needed to explore who is best placed to do this. Moreover, equity driven interventions could help to address socioeconomic disadvantaged (elderly) needs considering implementation of digital health literacy interventions.

## Supporting information

**S1 Checklist. Quality assessment (Mixed method appraisal tool checklist).**
(DOCX)

**S2 Checklist. PRISMA checklist.**
(DOCX)

**S1 Table. Search strategy.**
(DOCX)

**S2 Table. Number of participants of included studies.**
(DOCX)

**S3 Table. Target age group of included studies.**
(DOCX)

**S4 Table. Study design of included studies.**
(DOCX)

**S5 Table. Countries of included studies.**
(DOCX)

**S6 Table. Years of publication of included studies.**
(DOCX)

**S7 Table. Summary of Interventions description.**
(DOCX)

**S8 Table. Interventions measures, outcomes, effectiveness, and critique.**
(DOCX)

**S9 Table. Participants characteristics and the barriers and facilitators for intervention use by consumers.**
(DOCX)

## Author Contributions

**Conceptualization:** Hind Mohamed.

**Data curation:** Hind Mohamed.

**Formal analysis:** Hind Mohamed.

**Investigation:** Hind Mohamed, Esme Kittle, Nehal Nour, Ruba Hamed, Kaylem Feeney.

**Methodology:** Hind Mohamed.

**Project administration:** Hind Mohamed.

**Resources:** Hind Mohamed.

**Software:** Hind Mohamed.

**Supervision:** Jon Salsberg, Dervla Kelly.

**Validation:** Hind Mohamed, Esme Kittle.

**Visualization:** Hind Mohamed.

**Writing – original draft:** Hind Mohamed.

**Writing – review & editing:** Jon Salsberg, Dervla Kelly.

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
