## [Decision Letter · Decision Letter 0]

11 Jun 2024

PDIG-D-23-00445

An integrative systematic review on interventions to improve users’ ability to identify trustworthy digital health information.

PLOS Digital Health

Dear Dr. Mohamed,

Thank you for submitting your manuscript to PLOS Digital Health. After careful consideration, we feel that it has merit but does not fully meet PLOS Digital Health's publication criteria as it currently stands. Therefore, we invite you to submit a revised version of the manuscript that addresses the points raised during the review process.

Please submit your revised manuscript within 60 days Aug 10 2024 11:59PM. If you will need more time than this to complete your revisions, please reply to this message or contact the journal office at digitalhealth@plos.org. Please include the following items when submitting your revised manuscript:

We look forward to receiving your revised manuscript.

Kind regards,

Omar Badawi, Pharm.D., MPH

Section Editor

PLOS Digital Health

Journal Requirements:

Additional Editor Comments (if provided):

Reviewers' comments:

Reviewer's Responses to Questions

**Comments to the Author**

1. Does this manuscript meet PLOS Digital Health’s publication criteria? Is the manuscript technically sound, and do the data support the conclusions? The manuscript must describe methodologically and ethically rigorous research with conclusions that are appropriately drawn based on the data presented.

Reviewer #1: Yes

2. Has the statistical analysis been performed appropriately and rigorously?

Reviewer #1: Yes

3. Have the authors made all data underlying the findings in their manuscript fully available (please refer to the Data Availability Statement at the start of the manuscript PDF file)?

Reviewer #1: Yes

4. Is the manuscript presented in an intelligible fashion and written in standard English?

Reviewer #1: Yes

5. Review Comments to the Author

Reviewer #1: Please find below my comments on this study:

1. The title is clear but somewhat generic. Consider specifying the target audience or health context to make it more specific.

2. The introduction Identifies the significance of health literacy and digital health literacy in accessing reliable information. However, there are a few aspects that could be improved:

• The introduction lacks a specific and focused research gap or problem statement.

• Limited clarity on the primary objective or hypothesis of the study.

• The background section is broad and lacks specificity.

• The research questions could be more precise and directly tied to the objectives.

3. The methods section demonstrates a systematic and comprehensive approach to study selection, data extraction, quality assessment, and data synthesis. However, potential challenges arise from the inclusion of diverse study designs, multiple languages, and the broad study setting, which may affect the synthesis and generalizability of findings. Transparent reporting and addressing potential biases in the synthesis process are recommended.

4. The Results section provides an overview of the study's process, including the identification of records, screening, and inclusion/exclusion criteria. However, there are areas that could be improved: 

• The section on quality assessment could be clearer. Consider providing a concise summary of the key findings, focusing on methodological strengths and weaknesses of the included studies.

• The discussion of biases could be expanded to provide more insights into the potential impact on the study's findings and the overall quality of evidence.

• Consider concluding the Results section with a brief summary of the main findings, which could serve as a transition to the subsequent sections of the paper.

5. The discussion outlines the study's findings and attempts to draw meaningful conclusions. However, there are several aspects that could be criticized and improved:

• The use of subheadings within the discussion section could enhance the structure and make it easier for readers to navigate through different aspects of the study.

• Some points are repeated, such as the emphasis on the dominance of media literacy interventions. It's essential to avoid unnecessary repetition and focus on presenting new insights or reinforcing critical points more effectively.

• The discussion could benefit from a more critical evaluation of the quality and limitations of the studies reviewed. Discuss the methodologies used in the studies, potential biases, and the generalizability of the findings.

• There are instances where the findings are presented as contradictory, such as the disagreement with Car et al. (2011). Clarify the reasons for these inconsistencies and discuss potential factors contributing to differing results.

6. Conclusion:

• Could be more succinct and highlight the key takeaways from the study. Summarize the main findings and their implications for future research or practical applications.

• While the implications for practice are mentioned, they could be more explicit and actionable. Provide concrete recommendations for practitioners or policymakers based on the study's findings.

• Enhance the conclusion by proposing specific directions for future research based on the identified gaps and limitations.

Overall, the study offers valuable insights, but refining the organization, avoiding redundancy, and enhancing the title could contribute to an even stronger presentation of the research.

6. PLOS authors have the option to publish the peer review history of their article (what does this mean?). If published, this will include your full peer review and any attached files.

**Do you want your identity to be public for this peer review?** For information about this choice, including consent withdrawal, please see our Privacy Policy.

Reviewer #1: Yes: Y Zaki Almallah Clinical Professor of Urololg and Digital Health Cleveland Clinic Abu Dhabi

---

## [Editor Report · Decision Letter 1]

11 Sep 2024

An integrative systematic review on interventions to improve layperson’s ability to identify trustworthy digital health information.

PDIG-D-23-00445R1

Dear Dr. Mohamed,

We are pleased to inform you that your manuscript 'An integrative systematic review on interventions to improve layperson’s ability to identify trustworthy digital health information.' has been provisionally accepted for publication in PLOS Digital Health.

Best regards,

Jennifer N Avari Silva, MD

Section Editor

PLOS Digital Health

thank you for your response to previous review.